# Epileptic Encephalopathy *GABRB* Structural Variants Share Common Gating and Trafficking Defects

**DOI:** 10.3390/biom13121790

**Published:** 2023-12-14

**Authors:** Ciria C. Hernandez, Ningning Hu, Wangzhen Shen, Robert L. Macdonald

**Affiliations:** 1Life Sciences Institute, University of Michigan, Ann Arbor, MI 48109, USA; 2Department of Neurology, Vanderbilt University Medical Center, Nashville, TN 37232, USA; ningning.hu@vumc.org (N.H.); wangzhen.shen@vumc.org (W.S.); robert.macdonald@vumc.org (R.L.M.)

**Keywords:** *GABRB*, loss-of-function mutations, gain-of-function mutations, GABA_A_ receptors, stability and flexibility of GABA_A_ receptors, channel gating, receptor expression, epileptic encephalopathies, structure-function relationship, structural dynamics

## Abstract

Variants in the *GABRB* gene, which encodes the β subunit of the GABA_A_ receptor, have been implicated in various epileptic encephalopathies and related neurodevelopmental disorders such as Dravet syndrome and Angelman syndrome. These conditions are often associated with early-onset seizures, developmental regression, and cognitive impairments. The severity and specific features of these encephalopathies can differ based on the nature of the genetic variant and its impact on GABA_A_ receptor function. These variants can lead to dysfunction in GABA_A_ receptor-mediated inhibition, resulting in an imbalance between neuronal excitation and inhibition that contributes to the development of seizures. Here, 13 *de novo* EE-associated *GABRB* variants, occurring as missense mutations, were analyzed to determine their impact on protein stability and flexibility, channel function, and receptor biogenesis. Our results showed that all mutations studied significantly impact the protein structure, altering protein stability, flexibility, and function to varying degrees. Variants mapped to the GABA-binding domain, coupling zone, and pore domain significantly impact the protein structure, modifying the β+/α− interface of the receptor and altering channel activation and receptor trafficking. Our study proposes that the extent of loss or gain of GABA_A_ receptor function can be elucidated by identifying the specific structural domain impacted by mutation and assessing the variability in receptor structural dynamics. This paves the way for future studies to explore and uncover links between the incidence of a variant in the receptor topology and the severity of the related disease.

## 1. Introduction

Gamma-aminobutyric acid type A receptors (GABA_A_ receptors) are pentameric ligand-gated ion channels (pLGICs) composed of five subunits, 2α:2β:1γ, that come together to form a central pore [1,2,3,4,5,6,7,8]. Each GABA_A_ receptor subunit has four transmembrane domains (TM1–TM4), with a large extracellular domain between TM1 and TM2 (Figure 1A). The extracellular domain includes the N-terminal region and the Cys loop, which forms a disulfide bond between two cysteine residues. Upon binding of GABA to the extracellular domain, GABA_A_ receptors undergo a conformational change that leads to the opening of the ion channel. The conservation of the pentameric structure across GABA_A_ receptors contributes to the common gating signature shared by members of pLGICs [9,10,11,12]. The gating mechanism of GABA_A_ receptors is determined by structural motifs or “cassettes” that are conserved within this family. Structural studies that revealed the three-dimensional (3D) structure of GABA_A_ receptors, such as X-ray crystallography [9,10,11,12], cryo-electron microscopy [1,2,3,4,5,6,7,8], and computational methods [13,14,15], have provided valuable insights into the intricate structural cassettes that contribute to the gating of GABA_A_ receptors.

Mutations in *GABRB2* and *GABRB3* genes have been associated with certain forms of epileptic encephalopathies, such as Dravet syndrome, Ohtahara syndrome, and West syndrome [16,17,18,19,20,21,22,23,24,25,26]. Epileptic encephalopathies are a group of severe brain disorders characterized by the presence of seizures and abnormal electroencephalography (EEG) findings, often accompanied by cognitive and behavioral disturbances. They typically manifest in early childhood, leading to significant developmental delays or regression. Changes in *GABRB* function may lead to alterations in GABAergic inhibition, resulting in an imbalance between excitatory and inhibitory signals in the brain, which can contribute to epileptic activity. Mutations in structural cassettes that contribute to the gating of GABA_A_ receptors can significantly impact the channel’s function by altering the receptor’s conformation, ligand binding, protein stability, and flexibility, which may affect receptor function. These changes can lead to enhanced or diminished receptor function [17,19,21,22,23,24,25].

To gain insight into the molecular basis of how structural mutations in GABA_A_ receptor β2 and β3 subunits contribute to EEs, we chose 13 de novo *GABRB2* (M79T, F224C, F245S, I288S, V302M, K303N) and *GABRB3* (M80L, K127R, R232Q, Y245H, L278F, T281I, T287I) missense variants from the published literature reported in patients with a spectrum of EE phenotypes [16,18,20,22]. Using a combination of immunoblotting, automated patch clamp recording, and structural modeling, we characterized the effects of these *GABRB* mutations on GABA_A_ receptor dynamics, stability, flexibility, biogenesis, and channel function. We found that all these *GABRB2* and *GABRB3* mutations impaired GABA_A_ receptor biogenesis and channel function but to different extents. Furthermore, protein dynamics based on normal mode analysis predicted mutation-induced alterations in protein stability and flexibility upon amino acid substitution in the 3D structure of GABA_A_ receptors. Mutations that were found to have more significant effects consistent with a drastic reduction in receptor function were mapped along the β+/α− GABA-binding interface at the receptor’s N-terminal and transmembrane domains (TM), linked to gating and coupling. Structural perturbations at the site of the mutation were well correlated with defects in receptor gating and biogenesis, as the conformational changes resulting from vibrational entropy changes were mainly propagated through rearrangements through the β-strands and TM α-helices of the receptor.

Our studies highlight the usefulness of combining functional studies and molecular dynamics simulation to correlate protein structure and function. Understanding the effects of *GABRB* mutations on protein stability and flexibility is a novel area of research that analyzes the structural changes caused by pathogenic mutants in depth [27,28,29,30]. We demonstrated that *GABRB* mutations that led to significant changes in receptor function correlated with structural perturbations along the binding–coupling pathway. Research focusing on characterizing the functional and structural changes induced by *GABRB* mutations can provide valuable insights into developing targeted therapies for EEs.

## 2. Methods

### 2.1. 3D GABA_A_ Receptor Structures

The 3D structures of the α1β2γ2 heteropentameric GABA_A_ receptor (entry ID: 6X3X) (Figure 1A and Figure 2A) with a resolution of 2.92 Å [5] and the α1β3γ2 heteropentameric GABA_A_ receptor (entry ID: 6HUP) (Figure 3A) with a resolution of 3.58 Å [2] were collected in pdb format from the Protein Data Bank (rcsb.org). The chains’ IDs in the 3D structures are A, β2; B, α1; C, β2; D, α1; E, γ2 for 6X3X and B, β3; A, α1; E, β3; D, α1; C, γ2 for 6HUP.

### 2.2. Normal Mode Analysis (NMA): Analysis of Protein Dynamics

To analyze the protein structure–function relationship, we generated a consensus prediction of the GABA_A_ receptor dynamics (protein motions) that reflected the conformational repertories of the receptor by sampling conformation changes using DynaMut (biosig.unimelb.edu.au/dynamut), a structure-based computational approach [29]. DynaMut takes the 3D structure as an input to predict conformational changes using the NMA approach. Thus, the GABA_A_ receptor 3D structures 6X3X and 6HUP were submitted to a *Force Field C-Alpha* approach, and the calculations were performed for the first 10 non-trivial modes of the molecule to determine the *Deformation Energy* and *Atomic Fluctuation*. The deformation energy provided a measure of the local flexibility of the protein, while the atomic fluctuation provided the amplitude of the absolute atomic motion. The magnitude of the deformation/fluctuation of the structural domains of the receptor is represented by thin-to-thick tubes colored in blue (low), white (moderate), and red (high) in Figure 1B.

### 2.3. Correlation Analysis of the Stability and Flexibility of GABA_A_ Receptor Mutated Structures

Changes in the GABA_A_ receptor’s stability and flexibility upon single-point mutation were assessed using the DynaMut tool [29]. DynaMut implemented NMA by using EnCoM (Elastic Network Contact Model) and evaluated the impact of mutations on the protein’s dynamics and flexibility, resulting from vibrational entropy (*S*) changes between wild-type and mutant proteins (ΔΔ*S_Vib_ENCoM*). The resulting visual representation of vibrational entropy energy changes is shown as colored amino acids in the structural domain of the protein (Figure 4A and Figure 5A). Accordingly, *BLUE* represents a rigidification of the structure, and *RED* a gain in flexibility. For predicting the effects of the mutations on stability, we determined the difference in Gibbs free energy of folding (ΔΔ*G* = Δ*GWT* − Δ*GMT*, in Kcal/mol), with negative (ΔΔ*G* < 0) values denoting destabilizing mutations, and positive (ΔΔ*G* ≥ 0) values indicating stabilizing mutations (Figure 2B and Figure 3B). In addition, to analyze the impact of the mutations on membrane protein stability and the likelihood of them being disease-associated (*Pathogenicity*), DynaMut implemented an mCSM-membrane approach [30], where the mutations were labeled either as *Pathogenic* (F224C, F245S, I288S, K303N, K127R, Y245H, L278F, T281I, and T287I) or *Benign* (M79T, V302M, M80L, and R232Q).

### 2.4. DNA Constructs, Cell Culture, and Transfection of Human GABA_A_ Receptors

The coding sequences of the human α1 (*GABRA1*, NM_000806), β3 (*GABRB3*, NM_021912), β2 (GABRB2, NM_000813.3), and γ2 (*GABRG2*, NM_198904.2) GABA_A_R subunits were subcloned into the pcDNA3.1 expression vector (Invitrogen, Waltham, MA. USA). Mutant GABA_A_ receptor subunit constructs were generated using the QuikChange site-directed mutagenesis kit (Agilent Technologies, Santa Clara, CA, USA) and confirmed by DNA sequencing. Heterologous protein expression in transfected HEK293T cells (ATCC, CRL-11268) was described previously. In brief, for the expression experiments, wt and mutant α1β3γ2 and α1β2γ2eceptors, a total of 3 µg of α1, β3, β2, and γ2 (wt or mutant) subunit cDNAs was transfected at a ratio of 1:1:1 into cells plated in 60 mm culture dishes using polyethylenimine (PEI) (40 kD, Polysciences, Warrington, PA, USA). For automated patch clamp recordings, the cells were transfected after 24 h from plating with 3 μg of cDNA of each subunit (α1, β3, β2, and γ2, wild-type or mutant) using X-tremeGENE HP DNA transfection reagent (Roche Diagnostics, Indianapolis, IN, USA), following the manufacturer’s protocol. Recordings were obtained forty-eight hours after transfection.

### 2.5. Western Blot and Surface Biotinylation

The biotinylation protocol used was described previously [24]. The primary antibodies used to detect GABA_A_ receptors were as follows: mouse anti-α1 subunit antibody (1:500; NeuroMab, 75-136, Davis, CA, USA), rabbit anti-β3 subunit antibody (1:500; Novus, NB300-199, St. Louis, MO, USA), rabbit anti-β2 subunit antibody (1:1000; Millipore, AB5561, Burlington, MA, USA), and rabbit anti-γ2 subunit antibody (1:1000; Millipore, AB5559). The mouse anti-Na+/K+ ATPase antibody (1:1000; DSHB, a6F) was used to determine the band density of the loading control. IRDye^®^- (LI-COR Biosciences, Lincoln, NE, USA) conjugated secondary antibody was used at 1:10,000 dilution in all cases. The blotted membranes were scanned using the Odyssey Infrared Imaging System (LI-COR Biosciences). The Odyssey Image Studio software (LI-COR Biosciences, Image Studio^TM^ 5.0) selected the band-of-interest combined intensity values.

### 2.6. Determination of GABA-Elicited Responses by Automated Patch Clamp

Automated whole-cell patch clamp recordings were performed at room temperature on HEK293T cells 48 h after transfection with GABA_A_ receptor wt and mutant subunits, according to the manufacturer’s standard procedure for the SyncroPatch 384PE (Nanion Technologies, Munich, Germany), previously described [31,32]. Cells at a concentration of 400,000 cells/mL were harvested in suspension in an external solution containing 140 mM NaCl, 2 mM CaCl_2_, 4 mM KCl, 1 mM MgCl_2_, 5 mM glucose, and 10 mM HEPES (pH 7.4, adjusted with NaOH, 298 mOsm). Electrophysiological protocols integrated into the automated patch clamp system were employed, and the PatchControl 384 application by Nanion Technologies facilitated the digitization of the data. Following the establishment of a whole-cell configuration, the cells were perfused with an internal solution comprising 10 mM KCl, 10 mM NaCl, 110 mM KF, 10 mM EGTA, and 10 mM HEPES (pH 7.2, adjusted with KOH, 284 mOsm). Maintained at −80 mV throughout the experiment, GABA_A_ receptor currents were evoked by brief exposure (0.5 secs) to small GABA volumes (2 µL) at the respective experimental concentrations, utilizing the Ligand Puff function in PatchControl 384. Each experiment consisted of repeated GABA activation (three times), two washes between ligand applications to ensure complete washout, the application of a single ligand concentration per well, and full activation with 1 mM GABA (I_MAX_). For concentration–response (CRC) experiments, a single GABA concentration was applied to each well (ranging from 0.1 nM to 1 mM), and the maximum activation per well (cell) was determined using a subsequent maximum-effect (EC_MAX_) GABA concentration (1 mM). The currents were normalized to the maximal response of each well, and CRCs were calculated based on single-point additions. The EC_50_ values for each variant and the wild-type receptor were determined across multiple wells for each experimental condition. To calculate the EC_50_ values from the GABA CRCs, a three-parameter sigmoid model assuming a Hill slope of 1 (Eq 1: Y = Bottom + (Top-Bottom)/(1 + 10^((LogEC_50_ − X)))) was applied. The background reference signal (external solution) was subtracted, and the bottom plateau was constrained to zero. For the electrophysiological studies, we selected specific variants based on their location within the structural domain mapped to the β+/α− interface. The hypothesis was that homologous *GABRB* variants in this region would share common structural and activation defects. We prioritized variants that had not been previously studied functionally and considered variants whose surface expression was not significantly affected.

### 2.7. Statistical Analysis

Numerical data are reported as mean ± S.E.M. Statistical analyses were performed using GraphPad Prism (GraphPad Software 9.4).

## 3. Results

### 3.1. GABRB Mutations Were Mapped along the β+/α− GABA Binding Interface of GABA_A_ Receptors

α1β2/3γ2 GABA_A_ receptors comprise five subunits arranged symmetrically around a central ion channel pore (Figure 1A) [2,5]. Each subunit consists of four transmembrane domains (TMDs) labeled from M1 to M4. The M2 domain lines the ion channel pore and is crucial for ion selectivity and gating. The N-terminus of each subunit contains an extracellular ligand-binding domain (LBD). GABA binds to LBDs to specific sites at the interface between β+ and α− subunits, leading to conformational changes that open the ion channel pore. The existing knowledge on the transmission of motions from ligand-binding domains (LBDs) to TMDs and the gating motions of TMs is well accepted as describing common gating mechanisms through conserved structural cassettes shared among pLGICs [9,10,11,12]. Deformation energies and atomic fluctuation determined over the first 10 non-trivial conformational modes of the 6X3X structure of the pentameric α1β2γ2 GABA_A_ receptor revealed that the structural elements with the highest flexibility (ribbons in shades of red indicating the magnitude of local flexibility) were at the binding site and coupling zone of the receptor (Figure 1B, lower panels). Therefore, the NMA results demonstrated that the loops 2, A, B, F, C, the Cys loop, and the M2–M3 linker, positioned at the LBD–TM interface (coupling zone), are involved in propagating motions from the LBD to the TMDs and in activating GABA_A_ receptors (Figure 1B,C) [1,2,3,4,5,6].

### 3.2. GABRB Mutations Introduced Intra-Molecular Changes in the 3D Structure of GABA_A_ Receptors, Altering Receptor Stability

We hypothesized that missense mutations could disrupt protein stability by affecting intra-molecular interactions and introduce changes in the 3D structure of GABA_A_ receptors, affecting their ability to undergo conformational changes during ligand binding and channel opening. A comparison of the sequence alignments of *GABRB* genes (Figure 1C) confirmed that 13 de novo *GABRB2* (M79T, F224C, F245S, I288S, V302M, K303N) and *GABRB3* (M80L, K127R, Y245H, R232Q, L278F, T281I, T287I) mutations were mapped in the same structural elements crucial in the ligand binding–coupling mechanism of GABA_A_ receptors found by NMA (Figure 1B). To understand the impact of *GABRB* mutations on GABA_A_ receptor conformation, we thoroughly examined the interatomic interaction changes in the network of neighborhood residues upon mutations (Figure 2 and Figure 3). For predicting the effects of the mutations on receptor structure, the 3D 6X3X (Figure 2A) and 6HUP (Figure 3A) structures were chosen as representative members of pentameric α1β2γ2 and α1β3γ2 GABA_A_ receptors. The resulting predictions of *GABRB2* mutations indicated that M79T [M55T] in loop 2 lost interactions with residues in the Cys loop [L140] and the M2–M3 loop [P273], and F224C [F200C] in the C loop lost only one exchange interaction with a nearby residue in the Cys loop [E155]. In contrast, F245S [F221S] in M1 lost interactions with the Cys loop [Y143], pre-M1 loop [R216], and F loop [Q185] in the nearby α1 subunit. V302M [V278M] and K303N [K279N] in the M2–M3 loop and I288S [I264S] in M2 solely resulted in rearrangements of h-bonds among side chains of residues near the mutation (Figure 2B). Likewise, the homologous mutation in the β3 subunit, M80L [M55L] (Figure 3B), lost interactions with residues in the Cys loop [L140] and the M2–M3 loop [P273], but in contrast, kept intramolecular interactions with residues in loop 2 ([E52] and [N54]). K127R [K102R] in loop A lost h-bonds with residues in loop 2 ([N54] and [D56]), and R232Q [R207Q] in loop C lost interactions with [Y97] in loop A. Y245H [Y220H], facing the minus interface of β3, lost interactions with [Q224] in M1 and gained intramolecular interactions with residues in loop F ([P184] and [Q185]). L278F [L253F], T281I [T256I], and T287I [T262I] mutations in M2 resulted solely in rearrangements of the side chain of neighbor residues (Figure 3B). In general, the stability of a protein (receptor) is influenced by the interactions between amino acids, and mutations can disrupt these interactions, leading to protein misfolding or decreased stability. Thus, the structural perturbations caused by changes in the interatomic interactions of mutant residues affected the unfolding Gibbs free energy (ΔΔG, see methods), which measures the degree of alteration in GABA_A_ receptor stability after amino acid substitution. For comparison, the ΔΔG values related to each respective mutation are shown (Figure 2B and Figure 3B). All *GABRB* mutations, except two mutations (Y245H and T287I) in the β3 subunit, were predicted to decrease the receptor stability (negative ΔΔG values, shown in red). These results suggested that these mutations may interfere with proper protein folding, leading to the degradation of misfolded proteins by cellular quality control mechanisms. Disruptions in these interactions can impact the appropriate folding of the receptor subunits.

### 3.3. Loss- and Gain-of-Function Correlated with Increases and Decreases in GABA_A_ Receptor Flexibility

GABA_A_ receptors undergo conformational changes during ligand binding and channel opening. We hypothesized that *GABRB* mutations that alter receptor flexibility would potentially impact receptor function, as protein stability and flexibility changes can alter the sensitivity of GABA_A_ receptors to GABA. To determine GABA_A_ receptor conformational motions and the effect of *GABRB* mutations on receptor function, changes in the receptor flexibility were measured as the vibrational entropy difference (ΔS) between wild-type and mutant structures by implementing the ENCoM analysis (ΔΔS_Vib_ENCoM) (Figure 4A and Figure 5A). *GABRB* mutations resulted in significant alterations in the vibrational entropy of amino acids in the entire β subunit, with a consequent weighty gain in flexibility (ribbons colored in red) or stiffness (ribbons colored in blue). The ΔΔS_Vib_ENCoM values of each mutant receptor are shown (Figure 4A and Figure 5A). Therefore, *GABRB2* mutations decreased the receptor stability (negative ΔΔG values) (Figure 2B) by increasing the receptor flexibility (positive ΔΔS_Vib_ENCoM values) (Figure 4A). Similarly, five of the *GABRB3* mutations decreased the receptor stability (Figure 3B) by increasing the receptor flexibility (Figure 5A). In contrast, Y245H and T287I increased the receptor stability (positive ΔΔG values) (Figure 3B) by decreasing the receptor flexibility (negative ΔΔS_Vib_ENCoM values) (Figure 5A). These results suggested that one of the mechanisms by which these mutants affect GABA_A_ receptor activation may involve a restriction or facilitation in the transition of conformational motions during the coupling of ligand binding to channel opening, which would favor a specific conformational state.

Therefore, it is likely that these mutations disrupted the coupling of GABA binding to channel gating, leading to reduced or augmented GABA potency. We measured the effects of the β subunit mutations on GABA concentration–response curves (CRCs) (Figure 4B and Figure 5B, top panels). GABA_A_ receptor-evoked peak currents were obtained by applying various concentrations of GABA for 500 ms to wild-type α1β2/3γ2 and mutant GABA_A_ receptors. For wild-type α1β2γ2 GABA_A_ receptors, the EC_50_ for GABA-evoked currents was 8.99 µM (Figure 4B, black line). The GABA CRCs of the β2 (F224C), β2 (F245S), β2 (V302M), and β2 (K303N) subunits showed different magnitudes of GABA potency, with EC_50_ values of 4.87 µM (Figure 4B, orange line), 1.04 µM (Figure 4B, red line), 1.64 µM (Figure 4B, green line), and 0.15 µM (Figure 4B, blue line), respectively, which were left-shifted from 2- to 60-fold. For wild-type α1β3γ2 GABA_A_ receptors, the EC_50_ for GABA-evoked currents was 0.46 µM (Figure 5B, black line). The GABA CRCs of the β3 (M80L), β3 (K127R), β3 (Y245H), and β3 (L278F) subunits showed variations in the subunits’ sensitivity to GABA, with EC_50_ values of 19.5 µM (Figure 5B, blue line), 0.14 µM (Figure 5B, orange line), 0.17 µM (Figure 5B, green line), and 4.96 µM (Figure 5B, red line), respectively. Their EC_50_ values were either left-shifted 3-fold or right-shifted from 10- to 42-fold. Together, these results demonstrated that *GABRB2* mutations increased GABA potency (gain of function) by increasing the receptor flexibility (Figure 4A,B). Unexpectedly, *GABRB3* mutations behaved differently, reducing GABA potency (loss of function) by increasing the receptor flexibility (Figure 5A,B), while increasing GABA potency (gain of function) by either increasing or decreasing the receptor flexibility. The apparent lack of correlation between GABA potency and receptor flexibility caused by the *GABRB3* mutations may be explained by differences in the type of residue substituted (biochemical properties) and in the structural domain where the mutation occurred, which modified the function and structure of the receptor differently.

### 3.4. GABRB Mutations Altered the Surface and Total Levels of the α, β or γ Subunits, Which Correlated with GABA_A_ Receptor Stability

The proper folding of GABA_A_ receptor subunits is crucial for their functionality. We showed that most *GABRB* mutations were predicted to decrease receptor stability (negative ΔΔG values). To further investigate which type of GABA_A_ receptors were trafficked to the surface and whether the mutant β subunits had a dominant negative effect by decreasing the trafficking of the partnering subunits, we coexpressed α1 and γ2 subunits with wild-type or mutant β2 and β3 subunits. We analyzed the surface and total levels of GABA_A_ receptor subunits (Figure 4C,D and Figure 5C,D). When comparing the two total and surface expression data sets, all β2 mutations significantly affected all receptor subunits’ total and surface expression levels, but to different extents (Figure 4C,D). The impact of the β3 mutations was more prominent at the surface than at total expression levels (Figure 5C,D). Although these results seemed unrelated, we analyzed the possible correlation of the ΔΔG values determined by the mutant effect on receptor stability–folding (Figure 2B and Figure 3B) with the surface and total expression levels of each of the GABA_A_ receptor subunits (Figure 4B and Figure 5B, lower panels). Unexpectedly, we found a strong positive correlation of the impact of *GABRB2* mutations on receptor stability (ΔΔG) with the surface and total levels of expression of the α1, β2, and γ2 subunits (Figure 4B, lower panels). The Pearson correlation coefficient (*r*) values were 0.9665, 0.9997, and 0.9034 for surface expression and 0.9818, 0.9412, and 0.8634 for total expression. In the presence of *GABRB3* mutations, we observed a positive correlation of receptor stability (ΔΔG) with the surface expression levels and a negative correlation with the total expression levels of the α1, β3, and γ2 subunits (Figure 5B, lower panels). The Pearson correlation coefficient (*r*) values were 0.7507, 0.9824, and 0.9628 for surface expression and −0.9344, −0.2906, and −0.9415 for total expression.

## 4. Discussion

More precise and personalized treatment strategies are crucial for improving the management and outcomes of individuals with EEs associated with *GABRB* gene variants. Tailoring treatment approaches based on patients’ specific genetic profiles and clinical characteristics can lead to more effective interventions and better quality of life. Our studies highlight the multifaceted impact of *GABRB* missense mutations on GABA_A_ receptors, encompassing disruptions in protein stability, changes in the 3D structure, and effects on ligand binding and channel opening. Insights into these molecular consequences are crucial for elucidating the mechanisms through which genetic variations may contribute to functional changes in GABA_A_ [16,17,18,19,21,22,23,26].

Mutations in the *GABRB2* gene led to a gain-of-function effect in GABA_A_ receptors (Figure 2 and Figure 4). Specifically, these mutations resulted in an increased potency of GABA. This heightened potency implies that the mutated receptors exhibit a more robust response to GABA than the typical, unmutated receptors. The gain-of-function effect in this context was attributed to an increase in the flexibility of the GABA_A_ receptor. Enhanced receptor flexibility allowed for a more efficient and responsive interaction with GABA, increasing the potency of GABAergic signaling. This increased flexibility might alter the conformational dynamics of the receptor, influencing its ability to bind to GABA and transmit signals, ultimately resulting in a gain-of-function phenotype. In addition, there was a robust positive correlation between the impact of mutations in the *GABRB2* gene and the stability of the GABA_A_ receptor. This correlation was observed explicitly in terms of changes in free energy (ΔΔG) and was associated with both surface and overall expression levels of the α1, β2, and γ2 subunits of the GABA_A_ receptor (Figure 4). This impact was not only reflected in the changes in the free energy associated with these mutations (ΔΔG) but also evident in the levels of expression of the receptor subunits, both at the cell surface and in total. Our results suggest a connection between the genetic variations in *GABRB2*, the stability of the GABA_A_ receptor, and the abundance of specific receptor subunits.

The behavior of the *GABRB3* mutations appeared to be complex and diverse, involving both loss-of-function and gain-of-function effects on GABA_A_ receptors (Figure 3 and Figure 5). Mutations in the *GABRB3* gene resulted in a loss-of-function impact, as they reduced the potency of GABA. This implies that the mutated receptors exhibit a decreased response to GABA compared to the wild-type receptors. The loss-of-function effect was associated with increased flexibility of the GABA_A_ receptor. This heightened flexibility may disrupt the normal GABA-binding and signaling processes, reducing GABAergic transmission potency. Interestingly, despite the overall loss-of-function trend, a subset of GABRB3 mutations resulted in a gain-of-function effect. In this case, GABA potency was increased. This gain-of-function effect was associated with alterations in receptor flexibility. Importantly, it was noteworthy that both an increase and a decrease in receptor flexibility seemed to contribute to increased GABA potency. This indicates that the impact of *GABRB3* mutations on receptor flexibility can be bidirectional, leading to either enhanced or reduced GABAergic signaling efficacy. The dual effects of *GABRB3* mutations—reducing GABA potency through increased flexibility (loss of function) and increasing GABA potency through a variable impact on flexibility (gain of function)—highlights the intricate ways in which genetic variations can influence the function of GABA_A_ receptors [19,22,23,26]. Gain- and loss-of-function *GABRB3* variants have been reported to correlate with distinct clinical phenotypes in individuals with developmental and epileptic encephalopathies [22]. Nevertheless, a previous report that used concatenated oocytes showed significant differences in the functional characterization of *GABRB3* variants, particularly, the variants K127R and Y245H [22]. Our studies revealed that both variants, initially linked to either loss or gain of function, ultimately resulted in a gain of function. We used freely expressed α1, β3, and γ2 subunits in HEK cells, allowing for the examination of their independent assembly and trafficking to the cell surface. This approach contrasts with a prior report, which utilized a concatenated oocyte system. Heterologous systems, like HEK cells, offer distinct cellular contexts, assembly dynamics, and surface trafficking mechanisms that can significantly differ from those observed in native oocytes.

Moreover, mutations in the *GABRB3* gene exhibited distinct effects on the stability of the GABA_A_ receptor compared to *GABRB2* mutations. In the case of *GABRB3* mutations, there was a positive correlation with the stability of the GABA_A_ receptor, as indicated by changes in free energy (ΔΔG). This positive correlation was associated explicitly with the surface levels of the receptor. Interestingly, there was a negative correlation between the *GABRB3* mutations and the total levels of the α1, β3, and γ2 subunits of the GABA_A_ receptor. This suggests that these mutations might reduce the overall expression of these receptor subunits within the cell. In summary, *GABRB3* mutations seemed to influence GABA_A_ receptor stability positively, particularly at the cell surface, while simultaneously showing a negative correlation with the total expression levels of specific receptor subunits.

The flexibility of GABA_A_ receptors depends on the state of the receptor (open or closed) [1,2,4,6,8,12], which may explain why GABA potency does not always correlate with receptor flexibility resulting from *GABRB* mutations. The interplay between open and closed states is crucial for proper channel function. Mutations that induce increased flexibility in the open state may lead to shorter durations of the open state, which could impair channel opening. This is especially important in the coupling domain, which stabilizes intermediate states during receptor activation [1,2,4,7,8,13,15]. *GABRB* mutations may affect the delicate balance required for efficient channel gating, influencing GABA potency. The state-dependent nature of receptor flexibility adds complexity to our understanding, and the impact of mutations on GABA potency may be determined not only by the substituted residue but also by the state preferences introduced by the mutation.

Further investigations into the state-dependent aspects of flexibility are necessary to unravel the intricate interplay between structural modifications induced by mutations, receptor dynamics, and functional consequences to uncover the specific molecular mechanisms underlying these diverse effects of *GABRB*-associated EEs [16,17,18,20,22,23,33]. By integrating advancements in genetics, neuroscience, and personalized medicine, healthcare professionals can develop more targeted and effective treatment approaches for individuals with *GABRB*-associated epileptic encephalopathies [34].

## Figures and Tables

**Figure 1 biomolecules-13-01790-f001:**
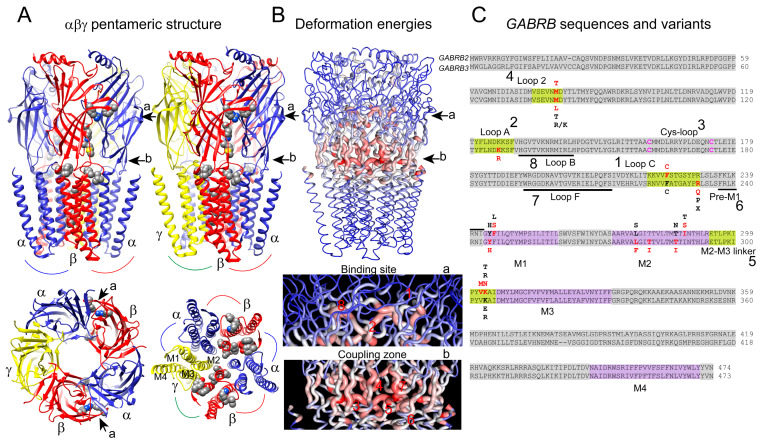
*GABRB*-associated EE mutations. (**A**) Cryo-EM structure of the pentameric α1β2γ2 GABA_A_ receptor (6X3X) viewed from the side and above, with the β subunits in red, the α subunits in blue, and the γ subunit in yellow. Arrows indicate the binding site (a) and the coupling zone (b). *GABRB* mutations are mapped onto the structure and represented as gray spheres. (**B**) Structural representation of deformation energies of the 6X3X structure. The magnitude of the deformation is represented by thin-to-thick tubes colored in blue (low), white (moderate), and red (high). Enlarged views of the deformation energies at the binding site (a-arrow) and the coupling zone (b-arrow) are shown in the lower panels. Structural domains part of the binding site and coupling zone are numbered and defined in panel (**C**). (**C**) The *GABRB* sequences with mapped missense mutations from this study are shown in bold red. The bold black color represents homologous missense mutations with different amino acid substitutions previously reported. Structural domains of the binding, coupling zone, and TMs are colored and described.

**Figure 2 biomolecules-13-01790-f002:**
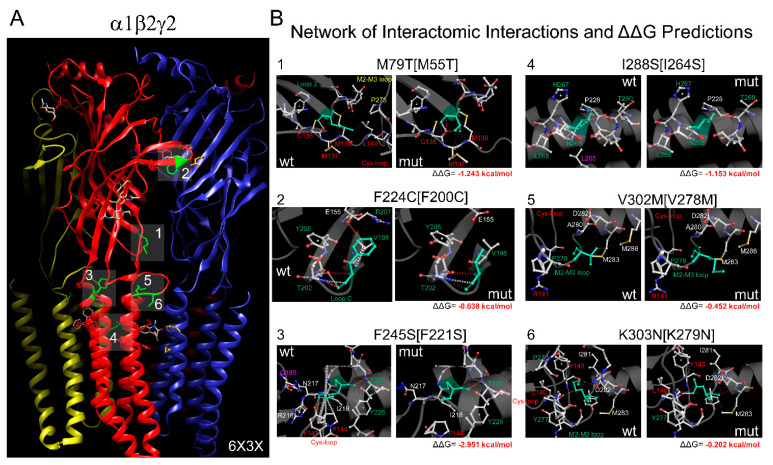
*GABRB2* mutations altered the stability of GABA_A_ receptors. (**A**) Cryo-EM structure of the pentameric α1β2γ2 GABA_A_ receptor (6X3X) viewed from the side, with the β subunits in red, the α subunits in blue, and the γ subunit in yellow. *GABRB2* mutations are mapped onto the structure and represented as green sticks and numbered according to the panels in (**B**). Network of interatomic interactions predicted by wild-type (wt, left panels) and β2 mutant (mut, right panels) 6X3X receptors. The panels are numbered according to the labels in panel (**A**), where the mutations are mapped. wt and mut amino acids are colored in light green, and the neighborhood residues are in CPK representation. All residues are represented as sticks. In brackets, the residues are numbered based on the 6X3X structure. For correspondence with the *GABRB2* protein sequence deposited in UniProt-P47870, 24 must be added to the number indicated on the panel. Structural domains involved in the network of interactions are labeled, and residues are equally colored. ΔΔG values are shown on the corresponding panels.

**Figure 3 biomolecules-13-01790-f003:**
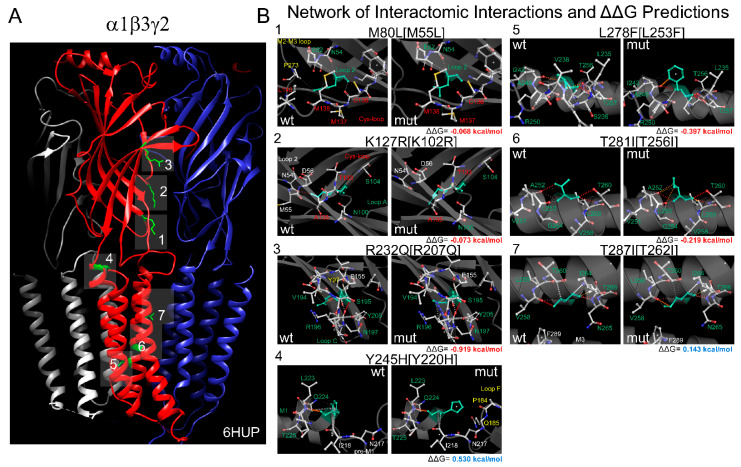
*GABRB3* mutations altered the stability of GABA_A_ receptors. (**A**) Cryo-EM structure of the pentameric α1β3γ2 GABA_A_ receptor (6HUP) viewed from the side, with the β subunits in red, the α subunits in blue, and the γ subunit in gray. *GABRB3* mutations are mapped onto the structure and represented as green sticks and numbered according to the panels in (**B**). Network of interatomic interactions predicted by wild-type (wt, left panels) and β3 mutant (mut, right panels) 6HUP receptors. The panels are numbered according to the labels in panel (**A**), where the mutations are mapped. wt and mut amino acids are colored in light green, and the neighborhood residues are in CPK representation. All residues are represented as sticks. In brackets, the residues are numbered based on the 6HUP structure. For correspondence with the *GABRB3* protein sequence deposited in UniProt-P28472, 25 must be added to the number indicated on the panel. Structural domains involved in the network of interactions are labeled, and residues are equally colored. ΔΔG values are shown on the corresponding panels.

**Figure 4 biomolecules-13-01790-f004:**
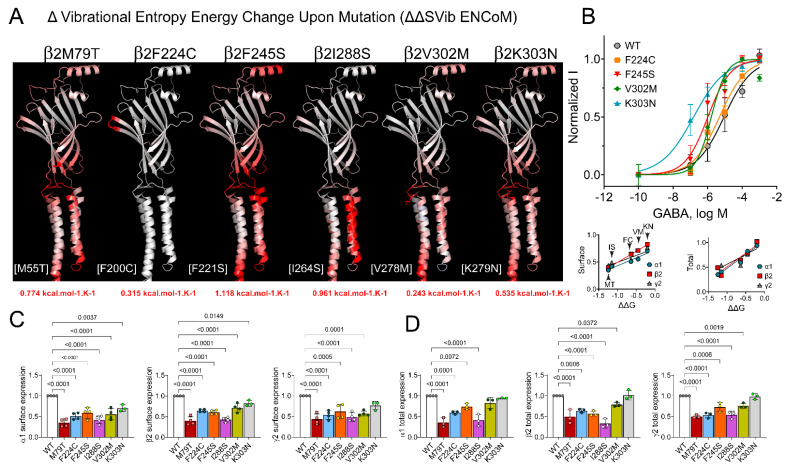
*GABRB2* mutations altered the flexibility and function of GABA_A_ receptors. (**A**) Structural representation of the GABA_A_ receptor flexible conformation based on the vibrational entropy energy change (∆∆SVib ENCoM) predicted between the wild-type and the mutant β2 subunit in the 6X3X structure. Mutant β2 subunits are colored according to the vibrational entropy perturbation upon mutation. Shades of blue represent a loss of flexibility (rigidification of the structure), and shades of red mean an increase in structural flexibility. ∆∆SVib ENCoM values are shown at the bottom of the corresponding mutation. In brackets, the variants are numbered based on the 6X3X structure. (**B**) In the upper panel, concentration–response curves of wild-type and mutant α1β2γ2 receptors are shown. GABA-evoked currents were normalized to the maximal response to 1 mM GABA. Lower panel, Pearson correlation coefficients obtained from the ∆∆G of the mutant β2 structures were plotted against surface (left graph) and total (right graph) expression levels of α1, β2, and γ2 subunits according to the data displayed in panels (**C**,**D**), respectively. Lines represent linear regression fits. (**C**,**D**) Surface and total expression levels of wild-type and mutant β2 subunits coexpressed with α1 and γ2 subunits, respectively.

**Figure 5 biomolecules-13-01790-f005:**
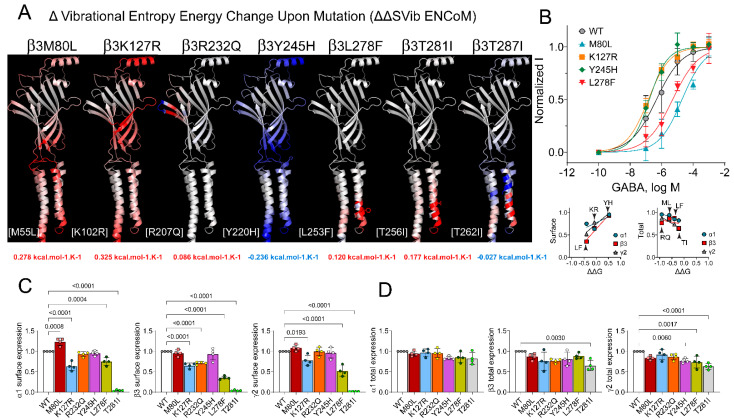
*GABRB3* mutations altered the flexibility and function of GABA_A_ receptors. (**A**) Structural representation of the GABA_A_ receptor flexible conformation based on the vibrational entropy energy change (∆∆SVib ENCoM) predicted between the wild-type and the mutant β3 subunit in the 6HUP structure. Mutant β3 subunits are colored according to the vibrational entropy perturbation upon mutation. Shades of blue represent a loss of flexibility (rigidification of the structure), and shades of red mean an increase in structural flexibility. ∆∆SVib ENCoM values are shown at the bottom of the corresponding mutation. In brackets, the variants are numbered based on the 6HUP structure. (**B**) In the upper panel, concentration–response curves of wild-type and mutant α1β3γ2 receptors are shown. GABA-evoked currents were normalized to the maximal response to 1 mM GABA. Lower panel, Pearson correlation coefficients obtained from the ∆∆G of the mutant β3 structures were plotted against surface (left graph) and total (right graph) expression levels of α1, β3, and γ2 subunits according to the data displayed in panels (**C**,**D**), respectively. Lines represent linear regression fits. (**C**,**D**) Surface and total expression levels of wild-type and mutant β3 subunits coexpressed with α1 and γ2 subunits, respectively.

## Data Availability

The current manuscript includes all the data supporting the reported results, involving datasets that were analyzed or generated during the study.

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
