# Peer review of "Epileptic Encephalopathy GABRB Structural Variants Share Common Gating and Trafficking Defects"

_biomolecules, 2023, doi:10.3390/biom13121790_

Round 1

Reviewer 1 Report

Comments and Suggestions for Authors

This paper investigates a correlation between the flexibility of the protein structure of GABAA receptor variants and the functional change. Although the variants are associated with different epilepsy types and severities, there is no attempt to draw any correlation between the flexibility and clinical features, making this a purely basic research paper. That is not a criticism, just a statement for clarity.

Overall, I think the idea of analysing differences in receptor structure to explain the behaviour of channel variants has merit and is worth investigating. For this specific manuscript, there are two few variants analyzed to draw anything but cautious conclusions from the data. No variants without a functional change were analyzed and the associations identified were limited to changes in expression rather than the GABA sensitivity that the structural regions are predominately involved in. While this may be useful in assessing clinical phenotypes from limited variant information in the future, that has not been evaluated here.

Major Issues

I think there needs to be some changes to the interpretation of the data. On the whole, there needs to be some sharpness to the conclusions that reflect that considerably more data is required for this to go from a speculative theory to a genuine association. For instance, the last part of the abstract states:

“Our study suggests that the degree of loss or gain of GABAA receptor function is explained by the structural domain affected by the mutation and the degree of variation in receptor structural dynamics, shedding light on the relationship between the variant topological occurrence and disease severity”.

There isn’t really anything in this study to suggest the structural domain is important, that is well reported elsewhere. The degree of change in function isn’t explained by receptor structural dynamics, but the results suggest these may play a part, or there may be an association between them. The manuscript would be greatly improved if the authors changed similar statements in the document to introduce greater clarity in how strong the association is, and make it clear that more work needs to be done.

 Minor Issues

Abstract: “were analyzed to determblotine” is nonsensical.

Should be mentioned that the change in function has been reported to correlate with different clinical phenotypes (ref 22), rather than merely different changes in function have been reported.

Introduction: “we identified 13 de novo GABRB2 (M79T, F224C, F245S, I288S, V302M, K303N) and GABRB3 (M80L, K127R, Y245H, R232Q, L278F, T281I, T287I) missense variants”

Better to state “we chose 13 de novo…from the published literature” to clarify that new patients are not being added.

Methods: “where mutations were labeled either as Pathogenic or Benign.” Which mutations were labelled as which and why? It wasn’t clear to me in the text.

The correlation between the free energy changes and the expression levels are probably the most important for the conclusion drawn in the manuscript, but it is presented in a very small panel. The authors should consider making this larger and more accessible.

There are differences between the results of this manuscript and others in the literature (e.g K127R). It’s not a major problem and shouldn’t obscure from the overall message of the publications, but the precise experiments are different and the difference could be stated briefly. 

In the results, the GABRB3 variants K127R and Y245H are different to those published in a different system (concatenated oocytes). The clinical descriptions of the K127R is consistent with a loss and Y245H a gain from the cohort data, so perhaps this should be stated in the discussion.

The different numbers on the protein structures to the bar graphs in the figures for the same variant is confusing. I suggest a standard notation and then the sequence of the pdb file in brackets for clarity.

In the discussion it states: "The apparent lack of correlation between GABA potency and receptor flexibility caused by the GABRB3 mutations may be explained by differences in the type of residue substituted (biochemical properties) and the structural domain where the mutation occurred, which will modify the function and structure of the receptor differently." 

I think this is a little simplistic. If flexibility is state-dependent, then either an open or closed state will be preferred (i.e. where the protein becomes more flexible in the open state it is likely to remain there for a shorter duration and impair channel opening). This is particularly so for the coupling region that stabilizes intermediate states. A statement should be incorporated explaining this.

Comments on the Quality of English Language

A few minor errors, just needs a thorough proofread.

Author Response

Reviewer 1

This paper investigates a correlation between the flexibility of the protein structure of GABAA receptor variants and the functional change. Although the variants are associated with different epilepsy types and severities, there is no attempt to draw any correlation between the flexibility and clinical features, making this a purely basic research paper. That is not a criticism, just a statement for clarity.

Overall, I think the idea of analysing differences in receptor structure to explain the behaviour of channel variants has merit and is worth investigating. For this specific manuscript, there are two few variants analyzed to draw anything but cautious conclusions from the data. No variants without a functional change were analyzed and the associations identified were limited to changes in expression rather than the GABA sensitivity that the structural regions are predominately involved in. While this may be useful in assessing clinical phenotypes from limited variant information in the future, that has not been evaluated here.

 We value the reviewer's feedback and have thoroughly examined our research in light of their comments. The manuscript has been enhanced and refined to incorporate the suggestions provided by the reviewer.

Major Issues

1-I think there needs to be some changes to the interpretation of the data. On the whole, there needs to be some sharpness to the conclusions that reflect that considerably more data is required for this to go from a speculative theory to a genuine association. For instance, the last part of the abstract states:

“Our study suggests that the degree of loss or gain of GABAA receptor function is explained by the structural domain affected by the mutation and the degree of variation in receptor structural dynamics, shedding light on the relationship between the variant topological occurrence and disease severity”.

 There isn’t really anything in this study to suggest the structural domain is important, that is well reported elsewhere. The degree of change in function isn’t explained by receptor structural dynamics, but the results suggest these may play a part, or there may be an association between them. The manuscript would be greatly improved if the authors changed similar statements in the document to introduce greater clarity in how strong the association is, and make it clear that more work needs to be done.

 R: We thank the reviewer for that thoughtful criticism. We agree to soften our conclusions to limit the scope of our research studies and leave future endeavors to cover the possible translational bench-to-clinic step. We rewrote the last paragraph of the abstract to make clear that change and in the discussion section.  

 Minor Issues

1-Abstract: “were analyzed to determblotine” is nonsensical.

R: The typo was corrected.

2-Should be mentioned that the change in function has been reported to correlate with different clinical phenotypes (ref 22), rather than merely different changes in function have been reported.

R: The modification has been implemented in line with the reviewer's suggestion. A paragraph has been introduced in the discussion section to address this change.

3-Introduction: “we identified 13 de novo GABRB2 (M79T, F224C, F245S, I288S, V302M, K303N) and GABRB3 (M80L, K127R, Y245H, R232Q, L278F, T281I, T287I) missense variants”

Better to state “we chose 13 de novo…from the published literature” to clarify that new patients are not being added.

R: This change has been addressed as suggested by the reviewer.

4-Methods: “where mutations were labeled either as Pathogenic or Benign.” Which mutations were labelled as which and why? It wasn’t clear to me in the text.

R: We apologize for the missing information. The classification of the variants has been added.

5-The correlation between the free energy changes and the expression levels are probably the most important for the conclusion drawn in the manuscript, but it is presented in a very small panel. The authors should consider making this larger and more accessible.

R: We appreciate the suggestion from the reviewer. However, limitations are imposed by the constraints on the number of figures and pages.

6-There are differences between the results of this manuscript and others in the literature (e.g K127R). It’s not a major problem and shouldn’t obscure from the overall message of the publications, but the precise experiments are different and the difference could be stated briefly. In the results, the GABRB3 variants K127R and Y245H are different to those published in a different system (concatenated oocytes). The clinical descriptions of the K127R is consistent with a loss and Y245H a gain from the cohort data, so perhaps this should be stated in the discussion.

R: We concur with the reviewer's feedback and regret the oversight regarding the missing information. As recommended, a paragraph addressing the observed differences has been incorporated into the discussion section.

7-The different numbers on the protein structures to the bar graphs in the figures for the same variant is confusing. I suggest a standard notation and then the sequence of the pdb file in brackets for clarity.

R: We thank the reviewer for the suggestion. The figures have been modified accordingly.

8-In the discussion it states: "The apparent lack of correlation between GABA potency and receptor flexibility caused by the GABRB3 mutations may be explained by differences in the type of residue substituted (biochemical properties) and the structural domain where the mutation occurred, which will modify the function and structure of the receptor differently." 

I think this is a little simplistic. If flexibility is state-dependent, then either an open or closed state will be preferred (i.e. where the protein becomes more flexible in the open state it is likely to remain there for a shorter duration and impair channel opening). This is particularly so for the coupling region that stabilizes intermediate states. A statement should be incorporated explaining this.

R: We appreciate the suggestion from the reviewer. A dedicated section has been crafted and included in the discussion.

Reviewer 2 Report

Comments and Suggestions for Authors

This manuscript describes 13 de novo missense mutations that have been found in the genes encoding the beta2 subunits (6 variants) and beta3 subunit (7 variants) of the GABA-A receptor.  There is apparently no formal evidence on a genome-wide basis that these mutations are truly associated with epileptic encephalopathy (EE), but based on what is known about GABA-A receptor-mediated inhibitory neurotransmission the assumption that these missense mutations significantly contribute to EE is very reasonable.  The authors do molecular modeling using cryo-EM structures, immunoblotting and automatic patch-clamping to characterize these mutations.  The mutations impact the protein structure, having effects on receptor stability, flexibility/rigidity and function.  The paper is well written.

Major points:

1)    One mutation in GABRB2, T287I, has not been included in the immunoblotting experiments in Fig. 5C,D.  What is the reason for this omission ?

2)    The patch-clamp experiments examining whether the mutants have an altered GABA sensitivity include only 4 out of the 6 GABRB2 mutations (Fig. 4B) and 4 out of the 7 GABRB3 mutations (Fig. 5B).  In a manuscript using an analysis pipeline that is not really novel, there should be no “missing data” unless there are very specific reasons why experiments could not be done (e.g., the specific missense mutants did not express in the recombinant system used).

Minor points:

1)    Abstract. Replace “determblotine” with “determine”.

2)    “6X3X structure”.  Please provide a reference the first time this structure is mentioned.

3)    Page 2 of 12: When listing the missense mutations in GABRB3, switch the positions of Y245H and R232Q to maintain the order.

4)    There are two numbering systems for GABA-A receptor subunits, one including and one excluding the signal peptide.  It does not make sense to use both numbering systems in the same figure in Figures 4 and 5, so that, e.g., the same mutation is called “M55T” in Fig. 4A and “M79T” in Fig. 4C,D or “M55L” in Fig. 5A and “M80L” in Fig. 5B,C,D.

5)    Page 7 of 12.  “1:10 000 dilution”.  Consider “1:10,000 dilution”.

6)    Page 8 of 12.  Please spell out “NMA” upon first occurrence.

7)    Page 9 of 12.  “shown different magnitude of GABA potency”.  This sentence should be reformulated.

Author Response

Reviewer 2

This manuscript describes 13 de novo missense mutations that have been found in the genes encoding the beta2 subunits (6 variants) and beta3 subunit (7 variants) of the GABA-A receptor.  There is apparently no formal evidence on a genome-wide basis that these mutations are truly associated with epileptic encephalopathy (EE), but based on what is known about GABA-A receptor-mediated inhibitory neurotransmission the assumption that these missense mutations significantly contribute to EE is very reasonable.  The authors do molecular modeling using cryo-EM structures, immunoblotting and automatic patch-clamping to characterize these mutations.  The mutations impact the protein structure, having effects on receptor stability, flexibility/rigidity and function.  The paper is well written.

 We express our gratitude for the thoughtful comments provided by the reviewer, and we have diligently revisited our studies. The manuscript has been enhanced in accordance with the valuable suggestions offered by the reviewer.

Major points:

  • One mutation in GABRB2, T287I, has not been included in the immunoblotting experiments in Fig. 5C,D.  What is the reason for this omission ?

R: We acknowledge the valuable comments from the reviewer. The GABRB3 T287I variant is positioned in a 2-α helix turn relative to the T281I variant within the M2 domain of β3. Structural modeling highlights highly similar structural perturbations at the mutation site, where both variants share the same substituted residue (Figures 3B and 5A). Critically, these mutations induce destabilization, resulting in improper folding and a lack of surface expression, while the total expression remains largely unaffected (data not presented for T287I). Notably, much like the T281I variant, the T287I variant exhibits markedly poor surface expression. As a result, we have chosen to present only one variant as a representative example.

  • The patch-clamp experiments examining whether the mutants have an altered GABA sensitivity include only 4 out of the 6 GABRB2 mutations (Fig. 4B) and 4 out of the 7 GABRB3 mutations (Fig. 5B).  In a manuscript using an analysis pipeline that is not really novel, there should be no “missing data” unless there are very specific reasons why experiments could not be done (e.g., the specific missense mutants did not express in the recombinant system used).

R: We appreciate the reviewers’ comments. In conducting electrophysiological studies, the choice of variants was contingent upon their positioning within the structural domain mapped to the β+/α- interface. This selection was guided by the hypothesis that homologous variants in this region would demonstrate common structural and activation defects. Additionally, priority was given to variants that had not been previously studied functionally, and lastly, consideration was given to variants whose surface expression was not significantly affected.

Therefore, as stated above, the GABRB2 variant I288S was previously reported to exhibit reduced maximum GABA-evoked currents and minor changes in EC50 in functional assessments (El Achkar et al, 2021). On a related note, M79T was identified as the inaugural de novo variant associated with epilepsy, positioned at the homologous site of the GABRB3 M80L variant. Both M79T and M80L are predicted to be deleterious and pathogenic. Notably, other GABRB3 variants at the M80 positions, with diverse substitutions (L, T, R, K), have been reported as loss-of-function mutations (Yang et al, 2021; Johannesen et al., 2021; Absalom et al., 2022), establishing this region as prone to variants linked to epilepsy. Our structural dynamics analysis confirmed that both M79T and M80L are destabilizing structural variants with comparable structural perturbations. Consequently, we anticipated that the gating effects would align with previous reports, especially given the similarity observed in the homologous variant GABRB3 M80T, classified as loss-of-function in Absalom et al. (2022). This expectation stems from the recognition that the structural domain governing the gating mechanism is equally affected in both M79T and M80L variants.

The surface expression of the GABRB3 variants T281I and T287I was notably poor, rendering the assessment of GABA concentration-response curves (GABA-CRCs) impractical. Furthermore, these variants exhibit shared structural domains, leading to comparable structural alterations. The GABRB3 R232Q variant, identified in an individual with Dravet syndrome (Le et al., 2017), has been categorized as a loss-of-function variant (Absalom et al., 2022). Positioned in Loop C, GABRB3 R232Q shares this region with GABRB2 F224C, and both variants converge on a common structural domain pivotal in channel gating. Variants located at Loop C and the b+/a- interface are anticipated to influence the GABA sensitivity of the receptor, as indicated by previous studies (Terejko et al., 2020; Nys et al., 2013; Bouzat, 2012).

Minor points:

  • Replace “determblotine” with “determine”.

R: The typo was corrected.

  • “6X3X structure”.  Please provide a reference the first time this structure is mentioned.

R: Reference was provided in the methods section.

  • Page 2 of 12: When listing the missense mutations in GABRB3, switch the positions of Y245H and R232Q to maintain the order.

R: The change of the order has been corrected.

  • There are two numbering systems for GABA-A receptor subunits, one including and one excluding the signal peptide.  It does not make sense to use both numbering systems in the same figure in Figures 4 and 5, so that, e.g., the same mutation is called “M55T” in Fig. 4A and “M79T” in Fig. 4C,D or “M55L” in Fig. 5A and “M80L” in Fig. 5B,C,D.

R: We apologize for the confusion in the variant numbering. As recommended by reviewer 1, we have implemented a standard notation and included the sequence of the pdb file in brackets for enhanced clarity.

  • Page 7 of 12.  “1:10 000 dilution”.  Consider “1:10,000 dilution”.

R: The change has been made as suggested.

  • Page 8 of 12.  Please spell out “NMA” upon first occurrence.

R: The abbreviation was initially defined upon its first appearance in the method section.

  • Page 9 of 12.  “shown different magnitude of GABA potency”.  This sentence should be reformulated.

R: The sentence has been revised in accordance with the suggestion.
